# Stateful Posted Pricing with Vanishing Regret via Dynamic Deterministic Markov Decision Processes

**Yuval Emek**
Technion - Israel Institute of Technology
Haifa, Israel
`yemek@technion.ac.il`

**Ron Lavi**
Technion - Israel Institute of Technology
Haifa, Israel
`ronlavi@ie.technion.ac.il`

**Rad Niazadeh**
University of Chicago Booth School of Business
Chicago, IL, United States
`rad.niazadeh@chicagobooth.edu`

**Yangguang Shi**
Technion - Israel Institute of Technology
Haifa, Israel
`shiyangguang@campus.technion.ac.il`

## Abstract

In this paper, a rather general online problem called *dynamic resource allocation with capacity constraints (DRACC)* is introduced and studied in the realm of posted price mechanisms. This problem subsumes several applications of stateful pricing, including but not limited to posted prices for online job scheduling and matching over a dynamic bipartite graph. As the existing online learning techniques do not yield vanishing-regret mechanisms for this problem, we develop a novel online learning framework defined over deterministic Markov decision processes with *dynamic* state transition and reward functions. We then prove that if the Markov decision process is guaranteed to admit an oracle that can simulate any given policy from any initial state with bounded loss — a condition that is satisfied in the DRACC problem — then the online learning problem can be solved with vanishing regret. Our proof technique is based on a reduction to online learning with *switching cost*, in which an online decision maker incurs an extra cost every time she switches from one arm to another. We formally demonstrate this connection and further show how DRACC can be used in our proposed applications of stateful pricing.

## 1 Introduction

Price posting is a common selling mechanism across various corners of e-commerce. Its applications span from more traditional domains such as selling flight tickets on Delta's website or selling products on Amazon, to more emerging domains such as selling cloud services on AWS or pricing ride-shares in Uber. The prevalence of price posting comes from its several important advantages: it is incentive compatible, simple to grasp, and can easily fit in an online (or dynamic) environment where buyers arrive sequentially over time. Therefore, online posted pricing mechanisms, also known as dynamic pricing, have been studied quite extensively in computer science, operations research, and economics (for a comprehensive survey, see [19]).

A very useful method for devising online posted prices is via *vanishing-regret online learning* algorithms in an adversarial environment [12, 11, 23, 9, 10, 28]. Here, a sequence of buyers arrive, each associated with her own valuation function that is assumed to be devised by a malicious adversary, and the goal is to post a sequence of price vectors that perform almost as good as the best fixed pricing policy in hindsight. Despite its success, a technical limitation of this method (shared by the aforementioned papers) forces the often less natural assumption of unlimited item supply to ensure that the selling platform is *stateless*. However, in many applications of online posted pricing,

the platform is *stateful*; indeed, prices can depend on previous sales that determine the platform's state. Examples for such stateful platforms include selling resources of limited supply, in which the state encodes the number of remaining inventories of different products, and selling resources in cloud computing to schedule online jobs, in which the state encodes the currently scheduled jobs.

The above mentioned limitation is in sharp contrast to the posted prices literature that consider stochastic settings where the buyers' valuations are drawn independently and identically from unknown distributions [8, 7, 35], or independently from known distributions [14, 22, 15]. By exploiting the randomness (and distributional knowledge) of the input and employing other algorithmic techniques, these papers cope with limited supply and occasionally, with more complicated stateful pricing scenarios. However, the stochastic approach does not encompass the (realistic) scenarios in which the buyers' valuations are correlated in various complex ways, scenarios that are typically handled using adversarial models. The only exception in this regard is the work of Chawla et al. [16] that takes a different approach: they consider the online job scheduling problem, and given access to a collection of (truthful) posted price scheduling mechanisms, they show how to design a (truthful) vanishing-regret online scheduling mechanism against this collection in an adversarial environment.

Motivated by the abundance of stateful posted pricing platforms, and inspired by [16], we study the design of adversarial online learning algorithms with vanishing regret for a rather general online resource allocation framework. In this framework, termed *dynamic resource allocation with capacity constraints (DRACC)*, dynamic resources of limited inventories arrive and depart over time, and an online mechanism sequentially posts price vectors to (myopically) strategic buyers with adversarially chosen combinatorial valuations (refer to Section 2 for the formal model). The goal is to post a sequence of price vectors with the objective of maximizing revenue, while respecting the inventory restrictions of dynamic resources for the periods of time in which they are active. We consider a full-information setting, in which the buyers' valuations are elicited by the platform after posting prices in each round of the online execution.

Given a collection of pricing policies for the DRACC framework, we aim to construct a sequence of price vectors that is guaranteed to admit a vanishing regret with respect to the best fixed pricing policy in hindsight. Interestingly, our abstract framework is general enough to admit, as special cases, two important applications of stateful posted pricing, namely, online job-scheduling and matching over a dynamic bipartite graph; these applications, for which existing online learning techniques fail to obtain vanishing regret, are discussed in detail in Appendix C.

**Our Contributions and Techniques.** Our main result is a vanishing-regret posted price mechanism for the DRACC problem (refer to Section 3 for a formal exposition).

> *For any DRACC instance with $T$ users and for any collection $\Gamma$ of pricing policies,*
> *the regret of our proposed posted price mechanism (in terms of expected revenue)*
> *with respect to the in-hindsight best policy in $\Gamma$ is sublinear in $T$.*

We prove this result by abstracting away the details of the pricing problem and considering a more general stateful decision making problem. To this end, we introduce a new framework, termed *dynamic deterministic Markov decision process (Dd-MDP)*, which generalizes the classic *deterministic MDP* problem to an adversarial online learning dynamic setting. In this framework, a decision maker picks a feasible action for the current state of the MDP, not knowing the state transitions and the rewards associated with each transition; the state transition function and rewards are then revealed. The goal of the decision maker is to pick a sequence of actions with the objective of maximizing her total reward. In particular, we look at vanishing-regret online learning, where the decision maker is aiming at minimizing her regret, defined with respect to the in-hindsight best fixed policy (i.e., a mapping from states to actions) among the policies in a given collection $\Gamma$.

Not surprisingly, vanishing-regret online learning is impossible for this general problem (see Proposition 3.1). To circumvent this difficulty, we introduce a structural condition on Dd-MDPs that enables online learning with vanishing regret. This structural condition ensures the existence of an ongoing *chasing oracle* that allows one to simulate a given fixed policy from any initial state, irrespective of the actual current state, while ensuring a small (vanishing) *chasing regret*. The crux of our technical contribution is cast in proving that the Dd-MDPs induced by DRACC instances satisfy this *chasability* condition.

Subject to the chasability condition, we establish a reduction from designing vanishing-regret online algorithms for Dd-MDP to the extensively studied (classic stateless) setting of *online learning with*

*switching cost* [26]. At high level, we have one arm for each policy in the given collection $\Gamma$ and employ the switching cost online algorithm to determine the next policy to pick. Each time this algorithm suggests a switch to a new policy $\gamma \in \Gamma$, we invoke the chasing oracle that attempts to simulate $\gamma$, starting from the current state of the algorithm which may differ from $\gamma$'s current state. In summary, we obtain the following result (see Theorem 3.8 for a formal exposition).

> *For any $T$-round Dd-MDP instance that satisfies the chasability condition and for any collection $\Gamma$ of policies, the regret of our online learning algorithm with respect to the in-hindsight best policy in $\Gamma$ is sublinear (and optimal) in $T$.*

We further study the bandit version of the above problem, where the state transition function is revealed at the end of each round, but the learner only observes the current realized reward instead of the complete reward function. By adapting the chasability condition to this setting, we obtain near optimal regret bounds. See Theorem E.2 and Corollary E.3 in Appendix E for a formal statement.

Our abstract frameworks, both for stateful decision making and stateful pricing, are rather general and we believe that they will turn out to capture many natural problems as special cases (on top of the applications discussed in Appendix C). The reader is referred to Appendix A for a more comprehensive discussion of the related literature.

## 2 Model and Definitions

**The DRACC problem.** Consider $N$ dynamic *resources* and $T$ strategic myopic *users* arriving sequentially over *rounds* $t = 1, \ldots, T$, where round $t$ lasts over the time interval $[t, t+1)$. Resource $i \in [N]$ arrives at the beginning of round $t_a(i)$ and departs at the end of round $t_e(i)$, where $1 \leq t_a(i) \leq t_e(i) \leq T$; upon arrival, it includes $c(i) \in \mathbb{Z}_{>0}$ units. We say that resource $i$ is *active* at time $t$ if $t_a(i) \leq t \leq t_e(i)$ and denote the set of resources active at time $t$ by $A_t \subseteq [N]$. Let $C$ and $W$ be upper bounds on $\max_{i \in [N]} c(i)$ and $\max_{t \in [T]} |A_t|$, respectively.

The arriving user at time $t$ has a *valuation function* $v_t : 2^{A_t} \to [0, 1)$ that determines her value $v_t(A)$ for each subset $A \subseteq A_t$ of resources active at time $t$. We assume that $v_t(\emptyset) = 0$ and that the users are quasi-linear, namely, if a subset $A$ of resources is allocated to user $t$ and she pays a total payment of $q$ in return, then her utility is $v_t(A) - q$. A family of valuation functions that receives a separated attention in this paper is that of $k_t$-*demand* valuation functions, where user $t$ is associated with an integer parameter $1 \leq k_t \leq |A_t|$ and with a value $w_t^i \in [0, 1)$ for each active resource $i \in A_t$ so that her value for a subset $A \subseteq A_t$ is $\max_{A' \subseteq A : |A'| \leq k_t} \sum_{i \in A'} w_t^i$.

**Stateful posted price mechanisms.** We restrict our attention to dynamic posted price mechanisms that work based on the following protocol. In each round $t \in [T]$, the mechanism first realizes which resources $i \in [N]$ arrive at the beginning of round $t$, together with their initial capacity $c(i)$, and which resources departed at the end of round $t - 1$, thus updating its knowledge of $A_t$. It then posts a *price vector* $\boldsymbol{p}_t \in (0, 1]^{A_t}$ that determines the price $\boldsymbol{p}_t(i)$ of each resource $i \in A_t$ at time $t$. Following that, the mechanism elicits the valuation function $v_t$ of the current user $t$ and allocates (or in other words sells) one unit of each resource in the demand set $\hat{A}_t^{\boldsymbol{p}_t}$ to user $t$ at a total price of $\hat{q}_t^{\boldsymbol{p}_t}$, where

$$\hat{A}_t^{\boldsymbol{p}} = \text{argmax}_{A \subseteq A_t} \left\{ v_t(A) - \sum_{i \in A} \boldsymbol{p}(i) \right\} \qquad \text{and} \qquad \hat{q}_t^{\boldsymbol{p}} = \sum_{i \in \hat{A}_t^{\boldsymbol{p}}} \boldsymbol{p}(i) \qquad (1)$$

for any price vector $\boldsymbol{p} \in (0, 1]^{A_t}$, consistently breaking $\text{argmax}$ ties according to the lexicographic order on $A_t$. A virtue of posted price mechanisms is that if the choice of $\boldsymbol{p}_t$ does not depend on $v_t$, then it is dominant strategy for (myopic) user $t$ to report her valuation $v_t$ truthfully.

Let $\boldsymbol{\lambda}_t \in \{0, 1, \ldots, C\}^{A_t}$ be the *inventory* vector that encodes the number $\boldsymbol{\lambda}_t(i)$ of units remaining from resource $i \in A_t$ at time $t = 1, \ldots, T$. Formally, if $t_a(i) = t$, then $\boldsymbol{\lambda}_t(i) = c(i)$; and if (a unit of) $i$ is allocated to user $t$ and $i$ is still active at time $t + 1$, then $\boldsymbol{\lambda}_{t+1}(i) = \boldsymbol{\lambda}_t(i) - 1$. We say that a price vector $\boldsymbol{p}$ is *feasible* for the inventory vector $\boldsymbol{\lambda}_t$ if $\boldsymbol{p}(i) = 1$ for every $i \in A_t$ such that $\boldsymbol{\lambda}_t(i) = 0$, that is, for every (active) resource $i$ exhausted by round $t$. To ensure that the resource inventory is not exceeded, we require that the posted price vector $\boldsymbol{p}_t$ is feasible for $\boldsymbol{\lambda}_t$ for every $1 \leq t \leq T$; indeed, since $v_t$ is always strictly smaller than 1, this requirement ensures that the utility of user $t$ from any resource subset $A \subseteq A_t$ that includes an exhausted resource is negative, thus preventing $A$ from becoming the selected demand set, recalling that the utility obtained by user $t$ from the empty set is 0.

In this paper, we aim for posted price mechanisms whose objective is to maximize the extracted *revenue* defined to be the total expected payment $\mathbb{E}[\sum_{t=1}^{T} \hat{q}_t^{\boldsymbol{p}_t}]$ received from all users, where the expectation is over the mechanism's internal randomness.[1]

**Adversarial online learning over pricing policies.**   To measure the quality of the aforementioned posted price mechanisms, we consider an adversarial online learning framework, where at each time $t \in [T]$, the decision maker picks the price vector $\boldsymbol{p}_t$ and an adaptive adversary simultaneously picks the valuation function $v_t$. The resource arrival times $t_a(i)$, departure times $t_e(i)$, and initial capacities $c(i)$ are also determined by the adversary. We consider the full information setting, where the valuation function $v_t$ of user $t$ is reported to the decision maker at the end of each round $t$. It is also assumed that the decision maker knows the parameters $C$ and $W$ upfront and that these parameters are independent of the instance length $T$.

A (feasible) *pricing policy* $\gamma$ is a function that maps each inventory vector $\boldsymbol{\lambda} \in \{0, 1, \ldots, C\}^{A_t}$, $t \in [T]$, to a price vector $\boldsymbol{p} = \gamma(\boldsymbol{\lambda})$, subject to the constraint that $\boldsymbol{p}$ is feasible for $\boldsymbol{\lambda}$.[2] The pricing policies are used as the benchmarks of our online learning framework: Given a pricing policy $\gamma$, consider a decision maker that repeatedly plays according to $\gamma$; namely, she posts the price vector $\boldsymbol{p}_t^{\gamma} = \gamma(\boldsymbol{\lambda}_t^{\gamma})$ at time $t = 1, \ldots, T$, where $\boldsymbol{\lambda}_t^{\gamma}$ is the inventory vector at time $t$ obtained by applying $\gamma$ recursively on previous inventory vectors $\boldsymbol{\lambda}_{t'}^{\gamma}$ and posting prices $\gamma(\boldsymbol{\lambda}_{t'}^{\gamma})$ at times $t' = 1, \ldots, t-1$. Denoting $\hat{q}_t^{\gamma} = \hat{q}_t^{\boldsymbol{p}_t^{\gamma}}$, the revenue of this decision maker is given by $\sum_{t=1}^{T} \hat{q}_t^{\gamma}$.

Now, consider a collection $\Gamma$ of pricing policies. The quality of a posted price mechanism $\{\boldsymbol{p}_t\}_{t=1}^{T}$ is measured by means of the decision maker's *regret* that compares her own revenue to the revenue generated by the in-hindsight best pricing policy in $\Gamma$. Formally, the regret (with respect to $\Gamma$) is defined to be

$$\max_{\gamma \in \Gamma} \sum_{t=1}^{T} \hat{q}_t^{\gamma} - \mathbb{E}\left[ \sum_{t=1}^{T} \hat{q}_t^{\boldsymbol{p}_t} \right],$$

where the expectation is taken over the decision maker's randomness. The mechanism is said to have *vanishing regret* if it is guaranteed that the decision maker's regret is sublinear in $T$, which means that the average regret per time unit vanishes as $T \to \infty$.

## 3   Dynamic Posted Pricing via Dd-MDP with Chasability

The online learning framework underlying the DRACC problem as defined in Section 2 is stateful with the inventory vector $\boldsymbol{\lambda}$ playing the role of the framework's state. In the current section, we first introduce a generalization of this online learning framework in the form of a stateful online decision making, formalized by means of *dynamic deterministic Markov decision processes (Dd-MDPs)*. Following that, we propose a structural condition called *chasability* and show that under this condition, the Dd-MDP problem is amenable to vanishing-regret online learning algorithms. This last result is obtained through a reduction to the extensively studied problem of "experts with switching cost" [26]. Finally, we prove that the Dd-MDP instances that correspond to the DRACC problem indeed satisfy the chasability condition.

### 3.1   Viewing DRACC as a Dd-MDP

A (static) *deterministic Markov decision process (d-MDP)* is defined over a set $\mathcal{S}$ of states and a set $\mathcal{X}$ of actions. Each state $s \in \mathcal{S}$ is associated with a subset $X_s \subseteq \mathcal{X}$ of actions called the feasible actions of $s$. A *state transition function* $g$ maps each state $s \in \mathcal{S}$ and action $x \in X_s$ to a state $g(s, x) \in \mathcal{S}$. This induces a directed graph over $\mathcal{S}$, termed the *state transition graph*, where an edge labeled by $\langle s, x \rangle$ leads from node $s$ to node $s'$ if and only if $g(s, x) = s'$. The d-MDP also includes a *reward function* $f$ that maps each state-action pair $\langle s, x \rangle$ with $s \in \mathcal{S}$ and $x \in X_s$ to a real value in $[0, 1]$.

**Dynamic deterministic MDPs.**   Notably, static d-MDPs are *not* rich enough to capture the dynamic aspects of the DRACC problem. We therefore introduce a more general object where the state transition and reward functions are allowed to develop in an (adversarial) dynamic fashion.

Consider a sequential game played between an online decision maker and an adversary. As in static d-MDPs, the game is defined over a set $\mathcal{S}$ of states, a set $\mathcal{X}$ of actions, and a feasible action set $X_s$ for each $s \in \mathcal{S}$. We further assume that the state and action sets are finite. The game is played in $T \in \mathbb{N}$ rounds as follows. The decision maker starts from an initial state $s_1 \in \mathcal{S}$. In each round $t = 1, \ldots, T$, she plays a (randomized) feasible action $x_t \in X_{s_t}$, where $s_t \in \mathcal{S}$ is the state at the beginning of round $t$. Simultaneously, the adversary selects the state transition function $g_t$ and the reward function $f_t$. The decision maker then moves to a new state $s_{t+1} = g_t(s_t, x_t)$ (which is viewed as a movement along edge $\langle s_t, x_t \rangle$ in the state transition graph induced by $g_t$), obtains a reward $f_t(s_t, x_t)$, and finally, observes $g_t$ and $f_t$ as the current round's (full information) feedback.[3] The game then advances to the next round $t + 1$. The goal is to maximize the expected total reward $\mathbb{E}[\sum_{t \in [T]} f_t(s_t, x_t)]$.

**Policies, simulation, & regret.** A (feasible) *policy* $\gamma : \mathcal{S} \mapsto \mathcal{X}$ is a function that maps each state $s \in \mathcal{S}$ to an action $\gamma(s) \in X_s$. A *simulation* of policy $\gamma$ over the round interval $[1, T]$ is given by the state sequence $\{s^\gamma(t)\}_{t=1}^T$ and the action sequence $\{x^\gamma(t)\}_{t=1}^T$ defined by setting

$$s^\gamma(t) \triangleq \begin{cases} s_1 & \text{if } t = 1 \\ g_{t-1}(s^\gamma(t-1), x^\gamma(t-1)) & \text{if } t > 1 \end{cases} \quad \text{and} \quad x^\gamma(t) \triangleq \begin{cases} \gamma(s_1) & \text{if } t = 1 \\ \gamma(s^\gamma(t)) & \text{if } t > 1 \end{cases}. \quad (2)$$

The cumulative reward obtained by this simulation of $\gamma$ is given by $\sum_{t \in [T]} f_t(s^\gamma(t), x^\gamma(t))$.

Consider a decision maker that plays the sequential game by following the (randomized) state sequence $\{s_t\}_{t=1}^T$ and action sequence $\{x_t\}_{t=1}^T$, where $x_t \in X_{s_t}$ for every $1 \le t \le T$. For a (finite) set $\Gamma$ of policies, the decision maker's *regret* with respect to $\Gamma$ is defined to be

$$\max_{\gamma \in \Gamma} \sum_{t \in [T]} f_t(s^\gamma(t), x^\gamma(t)) - \sum_{t \in [T]} \mathbb{E}[f_t(s_t, x_t)]. \quad (3)$$

### *Relation to the DRACC Problem*

Dynamic posted pricing for the DRACC problem can be modeled as a Dd-MDP. To this end, we identify the state set $\mathcal{S}$ with the set of possible inventory vectors $\boldsymbol{\lambda}_t$, $t = 1, \ldots, T$. If state $s \in \mathcal{S}$ is identified with inventory vector $\boldsymbol{\lambda}_t$, then we identify $X_s$ with the set of price vectors feasible for $\boldsymbol{\lambda}_t$. The reward function $f_t$ is defined by setting

$$f_t(s, x) = \hat{q}_t^x, \quad (4)$$

where $\hat{q}_t^x$ is defined as in Eq. (1), recalling that the valuation function $v_t$, required for the computation of $\hat{q}_t^x$, is available to the decision maker at the end of round $t$. As for the state transition function $g_t$, the new state $s' = g_t(s, x)$ is the inventory vector obtained by posting the price vector $x$ to user $t$ given the inventory vector $s$, namely,

$$s'(i) = \begin{cases} s(i) - 1_{i \in \hat{A}_t^x} & \text{if } i \in A_{t+1} \cap A_t \\ c(i) & \text{if } i \in A_{t+1} \setminus A_t \end{cases}.$$

Given the aforementioned definitions, the notion of (pricing) policies and their recursive simulations and the notion of regret translate directly from the DRACC setting to that of Dd-MDPs.

### 3.2 The Chasability Condition

As the Dd-MDP framework is very inclusive, it is not surprising that in general, it does not allow for vanishing regret (the proof of the following proposition is deferred to Appendix B).

**Proposition 3.1.** *For every online learning algorithm, there exists a $T$-round Dd-MDP instance for which the algorithm's regret is $\Omega(T)$.*

As a remedy to the impossibility result established in Proposition 3.1, we introduce a structural condition for Dd-MDPs that makes them amenable to online learning with vanishing regret.

**Definition 3.2** (*Chasability condition for Dd-MDPs*)**.** A Dd-MDP instance is called $\sigma$-*chasable* for some $\sigma > 0$ if it admits an ongoing *chasing oracle* $\mathcal{O}^{\text{Chasing}}$ that works as follows for any given target policy $\gamma \in \Gamma$. The chasing oracle is invoked at the beginning of some round $t_{\text{init}}$ and provided

with an initial state $s_{\text{init}} \in \mathcal{S}$; this invocation is halted at the end of some round $t_{\text{final}} \geq t_{\text{init}}$. In each round $t_{\text{init}} \leq t \leq t_{\text{final}}$, the chasing oracle generates a (random) action $\hat{x}(t)$ that is feasible for state

$$\hat{s}(t) = \begin{cases} s_{\text{init}} & \text{if } t = t_{\text{init}} \\ g_{t-1}\left(\hat{s}(t-1), \hat{x}(t-1)\right) & \text{if } t_{\text{init}} < t \leq t_{\text{final}} \end{cases} ; \qquad (5)$$

following that, the chasing oracle is provided with the Dd-MDP's state transition function $f_t(\cdot, \cdot)$ and reward function $g_t(\cdot, \cdot)$. The main guarantee of $\mathcal{O}^{\text{Chasing}}$ is that its *chasing regret (CR)* satisfies

$$\text{CR} \triangleq \sum_{t=t_{\text{init}}}^{t_{\text{final}}} f_t\left(s^\gamma(t), x^\gamma(t)\right) - \sum_{t=t_{\text{init}}}^{t_{\text{final}}} \mathbb{E}\left[f_t\left(\hat{s}(t), \hat{x}(t)\right)\right] \leq \sigma .$$

We emphasize that the initial state $s_{\text{init}}$ provided to the chasing oracle may differ from $s^\gamma(t_{\text{init}})$.

### *Relation to the DRACC Problem (continued)*

In terms of the DRACC problem, chasability means that the online algorithm can simulate a given pricing policy, while incurring a small revenue loss, even if the online algorithm starts from a (coordinate-wise) smaller inventory vector. Interestingly, the Dd-MDPs corresponding to DRACC instances are $\sigma$-chasable for $\sigma = o(T)$, where the exact bound on $\sigma$ depends on whether we consider general or $k_t$-demand valuation functions. Before establishing these bounds, we show that the chasing oracle must be randomized (proof deferred to Appendix B).

**Proposition 3.3.** *There exists a family of $T$-round DRACC instances whose corresponding Dd-MDPs do not admit a deterministic chasing oracle with $o(T)$ chasing regret CR.*

We now turn to study chasing oracles for DRACC instances implemented by randomized procedures.

**Theorem 3.4.** *The Dd-MDPs corresponding to $T$-round DRACC instances with $k_t$-demand valuation functions are $O(\sqrt{CW \cdot T})$-chasable.*

*Proof.* Consider some DRACC instance and fix the target pricing policy $\gamma \in \Gamma$; in what follows, we identify $\gamma$ with a decision maker that repeatedly plays according to $\gamma$. Given an initial round $t_{\text{init}}$ and an initial inventory vector $\hat{\boldsymbol{\lambda}}_{t_{\text{init}}}$, we construct a randomized chasing oracle $\mathcal{O}^{\text{Chasing}}$ that works as follows until it is halted at the end of round $t_{\text{final}} \geq t_{\text{init}}$. For each round $t_{\text{init}} \leq t \leq t_{\text{final}}$, recall that $\boldsymbol{\lambda}_t^\gamma$ is the inventory vector at time $t$ obtained by running $\gamma$ from round $1$ to $t$, and let $\hat{\boldsymbol{\lambda}}_t$ be the inventory vector at time $t$ obtained by $\mathcal{O}^{\text{Chasing}}$ as defined in Eq. (5). We partition the set $A_t$ of resources active at time $t$ into $\text{Good}_t = \{i \in A_t \mid \boldsymbol{\lambda}_t^\gamma(i) \leq \hat{\boldsymbol{\lambda}}_t(i)\}$ and $\text{Bad}_t = A_t \setminus \text{Good}_t$. In each round $t_{\text{init}} \leq t \leq t_{\text{final}}$, the chasing oracle posts the ($|A_t|$-dimensional) all-1 price vector with probability $\epsilon$, where $\epsilon \in (0, 1)$ is a parameter to be determined later on; and it posts the price vector

$$\hat{\boldsymbol{p}}_t = \begin{cases} \boldsymbol{p}_t^\gamma(i) & \text{if } i \in \text{Good}_t \\ 1 & \text{if } i \in \text{Bad}_t \end{cases}$$

with probability $1 - \epsilon$, observing that this price vector is feasible for $\hat{\boldsymbol{\lambda}}_t$ by the definition of $\text{Good}_t$ and $\text{Bad}_t$. Notice that $\mathcal{O}^{\text{Chasing}}$ never sells a resource $i \in \text{Bad}_t$ and that $\hat{\boldsymbol{p}}_t(i) \geq \boldsymbol{p}_t^\gamma(i)$ for all $i \in A_t$. Moreover, if resource $i$ arrives at time $t_a(i) = t > t_{\text{init}}$, then $i \in \text{Good}_t$.

To analyze the CR, we classify the rounds in $[t_{\text{init}}, t_{\text{final}}]$ into two classes called $\text{Following}$ and $\text{Missing}$: round $t$ is said to be $\text{Missing}$ if at least one (unit of a) resource in $\text{Bad}_t$ is sold by $\gamma$ in this round; otherwise, round $t$ is said to be $\text{Following}$. For each $\text{Following}$ round $t$, if $\mathcal{O}^{\text{Chasing}}$ posts $\hat{\boldsymbol{p}}_t$ in round $t$, then $\mathcal{O}^{\text{Chasing}}$ sells exactly the same resources as $\gamma$ for the exact same prices; otherwise ($\mathcal{O}^{\text{Chasing}}$ posts the all-1 price vector in round $t$), $\mathcal{O}^{\text{Chasing}}$ does not sell any resource. Hence, the CR increases in round $t$ by at most $\epsilon$ in expectation. For each $\text{Missing}$ round $t$, the CR increases in round $t$ by at most 1. Therefore the total CR over the interval $[t_{\text{init}}, t_{\text{final}}]$ is upper bounded by $\epsilon \cdot \mathbb{E}[\#\text{F}] + \mathbb{E}[\#\text{M}] \leq \epsilon \cdot T + \mathbb{E}[\#\text{M}]$, where $\#\text{F}$ and $\#\text{M}$ denote the number of $\text{Following}$ and $\text{Missing}$ rounds, respectively.

To bound $\mathbb{E}[\#\text{M}]$, we introduce a potential function $\phi(t)$, $t_{\text{init}} \leq t \leq t_{\text{final}}$, defined by setting

$$\phi(t) = \sum_{i \in \text{Bad}_t} \boldsymbol{\lambda}_t^\gamma(i) - \hat{\boldsymbol{\lambda}}_t(i)$$

By definition, $\phi(t_{\text{init}}) \leq CW$ and $\phi(t_{\text{final}}) \geq 0$. We argue that $\phi(t)$ is non-increasing in $t$. To this end, notice that if $t$ is a $\text{Following}$ round, then $\text{Bad}_{t+1} \subseteq \text{Bad}_t$, hence $\phi(t+1) \leq \phi(t)$. If $t$ is a $\text{Missing}$

round and $\mathcal{O}^{\text{Chasing}}$ posts the all-1 price vector, then $\phi(t+1) < \phi(t)$ as $\mathcal{O}^{\text{Chasing}}$ sells no resource whereas $\gamma$ sells at least one (unit of a) resource in $\text{Bad}_t$. So, it remains to consider a $\text{Missing}$ round $t$ in which $\mathcal{O}^{\text{Chasing}}$ posts the price vector $\hat{\boldsymbol{p}}_t$. Let $S^\gamma$ and $\hat{S}$ be the sets of (active) resources sold by $\gamma$ and $\mathcal{O}^{\text{Chasing}}$, respectively, in round $t$ and notice that a resource $i \in \hat{S} \setminus S^\gamma$ may move from $i \in \text{Good}_t$ to $i \in \text{Bad}_{t+1}$. The key observation now is that since $v_t$ is a $k_t$-demand valuation function, it follows that $S^\gamma \cap \text{Good}_t \subseteq \hat{S} \cap \text{Good}_t$, thus $|S^\gamma \cap \text{Bad}_t| \geq |\hat{S} \setminus S^\gamma|$. As both $\gamma$ and $\mathcal{O}^{\text{Chasing}}$ sell exactly one unit of each resource in $S^\gamma$ and $\hat{S}$, respectively, we conclude that $\phi(t+1) \leq \phi(t)$.

Therefore, $\mathbb{E}[\#\text{M}]$ is upper bounded by $CW$ plus the expected number of $\text{Missing}$ rounds in which $\phi(t)$ does not decrease. Since $\phi(t)$ strictly decreases in each $\text{Missing}$ round $t$ in which $\mathcal{O}^{\text{Chasing}}$ posts the all-1 price vector, it follows that the number of $\text{Missing}$ rounds in which $\phi(t)$ does not decrease is stochastically dominated by a negative binomial random variable $Z$ with parameters $CW$ and $\epsilon$. Recalling that $\mathbb{E}[Z] = (1 - \epsilon) \cdot CW/\epsilon$, we conclude that $\mathbb{E}[\#\text{M}] \leq CW + \mathbb{E}[Z] = CW/\epsilon$. The assertion is now established by setting $\epsilon = \sqrt{CW/T}$. $\qquad\square$

**Remark 3.5.** Theorem 3.4 can be in fact extended – using the exact same line of arguments – to a more general family of valuation functions $v_t$ defined as follows. Let $\boldsymbol{p}$ be a price vector, $B \subseteq A_t$ be a subset of the active resources, and $\boldsymbol{p}'$ be the price vector obtained from $\boldsymbol{p}$ by setting $\boldsymbol{p}'(i) = 1$ if $i \in B$; and $\boldsymbol{p}'(i) = \boldsymbol{p}(i)$ otherwise. Then, $|\hat{A}_t^{\boldsymbol{p}} \cap B| \geq |\hat{A}_t^{\boldsymbol{p}'} \setminus \hat{A}_t^{\boldsymbol{p}}|$. Besides $k_t$-demand valuations, this class of valuation functions includes OXS valuations [29] and single-minded valuations [30].

**Theorem 3.6.** *The Dd-MDPs corresponding to $T$-round DRACC instances with arbitrary valuation functions are $O\left(T^{\frac{CW}{CW+1}}\right)$-chasable.*

*Proof.* The proof follows the same line of arguments as that of Theorem 3.4, only that now, it no longer holds that the potential function $\phi(t)$ is non-increasing in $t$. However, it is still true that (I) $0 \leq \phi(t) \leq CW$ for every $t_{\text{init}} \leq t \leq t_{\text{final}}$; (II) if $t_{\text{init}} \leq t < t_{\text{final}}$ is a $\text{Missing}$ round and $\mathcal{O}^{\text{Chasing}}$ posts the all-1 price vector in round $t$, then $\phi(t+1) < \phi(t)$; and (III) if $\phi(t) = 0$ for some $t_{\text{init}} \leq t \leq t_{\text{final}}$, then $\phi(t') = 0$ for all $t < t' \leq t_{\text{final}}$. We conclude that if $\mathcal{O}^{\text{Chasing}}$ posts the all-1 price vector in $CW$ contiguous $\text{Missing}$ rounds, then $\phi(\cdot)$ must reach zero and following that, there are no more $\text{Missing}$ rounds. Therefore the total number $\#\text{M}$ of $\text{Missing}$ rounds is stochastically dominated by $CW$ times a geometric random variable $Z$ with parameter $\epsilon^{CW}$. Since $\mathbb{E}[Z] = \epsilon^{-CW}$, it follows that $\mathbb{E}[\#\text{M}] \leq CW/\epsilon^{CW}$. Combined with the $\text{Following}$ rounds, the CR is upper bounded by $\epsilon \cdot T + CW/\epsilon^{CW}$. The assertion is established by setting $\epsilon = (T/(CW))^{-1/(CW+1)}$. $\qquad\square$

### 3.3 Putting the Pieces Together: Reduction to Online Learning with Switching Cost

Having an ongoing chasing oracle with vanishing chasing regret in hand, our remaining key technical idea is to reduce online decision making for the Dd-MDP problem to the well-studied problem of *online learning with switching cost (OLSC)* [26]. The problem's setup under full-information is exactly the same as the classic problem of learning from experts' advice, but the learner incurs an extra cost $\Delta > 0$, a parameter referred to as the *switching cost*, whenever it switches from one expert to another. Here, we have a finite set $\Gamma$ of experts (often called actions or arms) and $T \in \mathbb{Z}_{>0}$ rounds. The expert reward function $F_t : \Gamma \mapsto [0, 1)$ is revealed as feedback at the end of round $t = 1, \ldots, T$. The goal of an algorithm for this problem is to pick a sequence $\gamma_1, \ldots, \gamma_T$ of experts in an online fashion with the objective of minimizing the regret, now defined to be

$$\max_{\gamma \in \Gamma} \sum_{t \in [T]} F_t(\gamma) - \left( \sum_{t \in [T]} \mathbb{E}\left[ F_t(\gamma_t) \right] - \Delta \cdot \sum_{t=2}^{T} \mathbb{1}_{\gamma_t \neq \gamma_{t-1}} \right) .$$

**Theorem 3.7** ([26])**.** *The OLSC problem with switching cost $\Delta$ admits an online algorithm $\mathcal{A}$ whose regret is $O\left(\sqrt{\Delta \cdot T \log |\Gamma|}\right)$.*

Note that the same theorem also holds for independent stochastic switching costs with $\Delta$ as the upper bound on the expected switching cost, simply because of linearity of expectation and the fact that in algorithms for OLSC, such as the Following-The-Perturbed-Leader [26], switching at each time is independent of the realized cost of switching.

We now present our full-information online learning algorithm for $\sigma$-chasable Dd-MDP instances; the reader is referred to Appendix E for the bandit version of this algorithm. Our (full-information)

---
**ALGORITHM 1:** Online Dd-MDP algorithm C&S
---
**Input:** Policy set $\Gamma$, OLSC algorithm $\mathcal{A}$, chasing oracle $\mathcal{O}^{\texttt{Chasing}}$, initial state $s_1$;
**Output:** Sequence $x_1, \ldots, x_T$ of actions, (implicit) sequence $s_2, \ldots, s_T$ of states;

Start from initial state $s_1$;
**for** *each round $t \in [T]$* **do**
    Invoke $\mathcal{A}$ to pick a policy $\gamma_t$ at the beginning of round $t$;
    **if** *$t > 1$ and $\gamma_t \neq \gamma_{t-1}$* **then**
        Invoke $\mathcal{O}^{\texttt{Chasing}}$ from scratch with target policy $\gamma_t$, initialized with round $t$ and state $s_t$;
        Select the action $x_t \leftarrow \hat{x}(t)$ returned by $\mathcal{O}^{\texttt{Chasing}}$;
    **else**
        Continue the existing run of $\mathcal{O}^{\texttt{Chasing}}$ and select the action $x_t \leftarrow \hat{x}(t)$ it returns;
    Feed $\mathcal{O}^{\texttt{Chasing}}$ with $g_t(\cdot, \cdot)$ and $f_t(\cdot, \cdot)$ as the state transition and reward functions of round $t$;
    **for** *each $\gamma \in \Gamma$* **do**
        Compute $F_t(\gamma) \leftarrow f_t(s^\gamma(t), x^\gamma(t))$ by simulating policy $\gamma$ up to time $t$ (see Eq. (2));
    Feed $\mathcal{A}$ with $F_t(\cdot)$ as the reward function of round $t$;
---

algorithm, called *chasing and switching (C&S)*, requires a black box access to an algorithm $\mathcal{A}$ for the OLSC problem with the following configuration: (1) the expert set of $\mathcal{A}$ is identified with the policy collection $\Gamma$ of the Dd-MDP instance; (2) the number of rounds of $\mathcal{A}$ is equal to the number of rounds of the Dd-MDP instance ($T$); and (3) the switching cost of $\mathcal{A}$ is set to $\Delta = \sigma$.

The operation of C&S is described in Algorithm 1. This algorithm maintains, in parallel, the OLSC algorithm $\mathcal{A}$ and an ongoing chasing oracle $\mathcal{O}^{\texttt{Chasing}}$; $\mathcal{A}$ produces a sequence $\{\gamma_t\}_{t=1}^T$ of policies and $\mathcal{O}^{\texttt{Chasing}}$ produces a sequence $\{x_t\}_{t=1}^T$ of actions based on that. Specifically, $\mathcal{O}^{\texttt{Chasing}}$ is *restarted*, i.e., invoked from scratch with a fresh policy $\gamma$, whenever $\mathcal{A}$ switches to $\gamma$ from some policy $\gamma' \neq \gamma$.

**Theorem 3.8.** *The regret of C&S for $T$-round $\sigma$-chasable Dd-MDP instances is $O\left(\sqrt{\sigma \cdot T \log |\Gamma|}\right)$.*

*Proof.* Partition the $T$ rounds into episodes $\{1, 2, \ldots\}$ so that each episode $\theta$ is a maximal contiguous sequence of rounds in which the policy $\gamma_\theta$ chosen by $\mathcal{A}$ does not change. Let $t_\theta$ and $t'_\theta$ be the first and last rounds of episode $\theta$, respectively. Consider some episode $\theta$ with corresponding policy $\gamma_\theta$. Since C&S follows an action sequence generated by $\mathcal{O}^{\texttt{Chasing}}$ during the round interval $[t_\theta, t'_\theta]$ and since the chasing regret of $\mathcal{O}^{\texttt{Chasing}}$ is upper bounded by $\sigma = \Delta$, it follows that

$$\sum_{t=t_\theta}^{t'_\theta} F_t(\gamma_\theta) - \sum_{t=t_\theta}^{t'_\theta} \mathbb{E}\left[f_t(s_t, x_t)\right] = \sum_{t=t_\theta}^{t'_\theta} f_t\left(s^{\gamma_\theta}(t), x^{\gamma_\theta}(t)\right) - \sum_{t=t_\theta}^{t'_\theta} \mathbb{E}\left[f_t(s_t, x_t)\right] \leq \Delta.$$

Therefore, for each policy $\gamma \in \Gamma$, we have

$$\sum_{t \in [T]} f_t\left(s^\gamma(t), x^\gamma(t)\right) - \sum_{t \in [T]} \mathbb{E}\left[f_t(s_t, x_t)\right] \leq \sum_{t \in [T]} f_t\left(s^\gamma(t), x^\gamma(t)\right) - \sum_\theta \left(\sum_{t=t_\theta}^{t'_\theta} \mathbb{E}\left[F_t(\gamma_\theta)\right] - \Delta\right)$$

$$= \sum_{t \in [T]} F_t(\gamma) - \left(\sum_{t \in [T]} \mathbb{E}\left[F_t(\gamma_t)\right] - \Delta \cdot \sum_{t=2}^T 1_{\gamma_t \neq \gamma_{t-1}}\right).$$

By Theorem 3.7, the last expression is at most $O\left(\sqrt{\Delta \cdot T \log |\Gamma|}\right) = O\left(\sqrt{\sigma \cdot T \log |\Gamma|}\right)$. $\qquad\square$

So far, we have only considered the notion of *policy regret* as defined in Eq. (3). An extension of our results to the notion of *external regret* [4] is discussed in Appendix D. Furthermore, we investigate the bandit version of the problem in Appendix E. In a nutshell, by introducing a stateless version of our full-information chasing oracle and reducing to the adversarial multi-armed-bandit problem [5], we obtain $O(T^{2/3})$ regret bound for Dd-MDP under bandit feedback. Finally, we obtain near-matching lower bounds for both the full-information and bandit feedback versions of the Dd-MDP problem under the chasability condition in Appendix F.

***Relation to the DRACC Problem (continued)***

We can now use C&S (Algorithm 1) for Dd-MDPs that correspond to DRACC instances. This final mechanism is called *learning based posted pricing (LBPP)*. It first provides the input parameters of C&S, including the collection $\Gamma$ of pricing policies, the OLSC algorithm $\mathcal{A}$, the ongoing chasing oracle $\mathcal{O}^{\text{Chasing}}$ and the initial state $s_1$. It then runs C&S by posting its price vectors (actions) and updating the resulting inventory vectors (states). For $\mathcal{O}^{\text{Chasing}}$, we employ the (randomized) chasing oracles promised in Theorem 3.4 and Theorem 3.6. The following theorems can now be inferred from Theorem 3.8, Theorem 3.4, and Theorem 3.6.

**Theorem 3.9.** *The regret of* LBPP *for $T$-round DRACC instances with $k_t$-demand valuation functions (or more generally, with the valuation functions defined in Remark 3.5) is $O\left((CW)^{\frac{1}{4}}T^{\frac{3}{4}}\sqrt{\log|\Gamma|}\right)$.*

**Theorem 3.10.** *The regret of* LBPP *for $T$-round DRACC instances with with arbitrary valuation functions is $O\left(T^{\frac{1}{2}\left(1+\frac{CW}{CW+1}\right)}\sqrt{\log|\Gamma|}\right)$.*

Note that the regret bounds in Theorem 3.9 and Theorem 3.10 depend on the parameters $C$ and $W$ of the DRACC problem; as shown in Appendix G, such a dependence is unavoidable.

## Broader Impact

The current paper presents theoretical work without any foreseeable societal consequence. Therefore, the authors believe that the broader impact discussion is not applicable.

## Acknowledgments and Disclosure of Funding

The work of Yuval Emek was supported in part by an Israel Science Foundation grant number 1016/17.

The work of Ron Lavi was partially supported by the Israel Science Foundation - Natural Science Foundation of China joint research program (grant No. 2560/17).

The work of Yangguang Shi was partially supported at the Technion by a fellowship of the Israel Council for Higher Education.

Funding in direct support of this work: ISF grant number 1016/17, ISF-NSFC grant No. 2560/17, and a fellowship of the Israel Council for Higher Education.

## Footnotes

[1]The techniques we use in this paper are applicable also to the objective of maximizing the social welfare.

[2]The seemingly more general setup, where the time $t$ is passed as an argument to $\gamma$ on top of $\boldsymbol{\lambda}$, can be easily reduced to our setup (e.g., by introducing a dummy resource $i_t$ active only in round $t$).

[3] No (time-wise) connectivity assumptions are made for the dynamic transition graph induced by $\{g_t\}_{t=1}^T$, hence it may not be possible to devise a path between two given states as is done in [17] for static d-MDPs.

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
