[Supplementary Material]

# APPENDIX

## A  Further Related Work

**Additional related Work and discussions.**    In the DRACC problem, the class of feasible prices at each time $t$ is determined by the remaining inventories, which in turn depends on the prices picked at previous times $t' < t$. This kind of dependency cannot be handled by the conventional online learning algorithms, such as follow-the-perturbed-leader [26] and EXP3 [6]. That is why we aim for the stateful model of online learning, which allows a certain degree of dependence on the past actions.

Several attempts have been made to formalize and study stateful online learning models. The authors of [3, 23] consider an online learning framework where the reward (or cost) at each time depends on the $k$ recent actions for some fixed $k > 0$. This framework can be viewed as a reward function that depends on the system's state that, in this case, encodes the last $k - 1$ actions.

There is an extensive line of work on online learning models that address general multi-state systems, typically formalized by means of stochastic [21, 25, 34, 1, 32] or deterministic [17] MDPs. The disadvantage of these models from our perspective is that they all have at least one of the following two restrictions: (a) all actions are always feasible regardless of the current state [1, 21, 25, 34]; or (b) the state transition function is fixed (static) and known in advance [17, 21, 25, 32, 34].

In the DRACC problem, however, not all actions (price vectors) are feasible for every state and the state transition function at time $t$ is revealed only after the decision maker has committed to its action. Moreover, the aforementioned MDP-based models require a certain type of state *connectivity* in the sense that the Markov chain induced by each action should be irreducible [1, 21, 25, 32, 34] or at least the union of all induced Markov chains should form a strongly connected graph [17]. In contrast, in the DRACC problem, depending on the inventories of the resources, it may be the case that certain inventory vectors can never be reached (regardless of the decision maker's actions).

On the algorithmic side, a common feature of all aforementioned online learning models is that for every instance, there exists some $k > 0$ that can be computed in a preprocessing stage (and does not depend on $T$) such that the online learning can "catch" the state (or distribution over states) of any given sequence of actions in *exactly* $k$ time units. While this feature serves as a corner stone for the existing online learning algorithms, it is not present in our model, hence our online learning algorithm has to employ different ideas.

In [20, 27, 2], a family of online resource allocation problems is investigated under a different setting from ours. The resources in their problem models are static, which means that every resource is revealed at the beginning, and remains active from the first user to the last one. Different from our adversarial model, these papers take different stochastic settings on the users, such as the random permutation setting where a fixed set of users arrive in a random order [27, 2], and the random generation setting where the parameters of each user are drawn from some distribution [20, 2]. In these papers, the assignment of the resources to the requests are fully determined by a single decision maker, and the decision for each request depends on the revealed parameters of the current request and previous ones. By contrast, we study the scenario where each strategic user makes her own decision of choosing the resources, and the price posted to each user should be specified independently of the valuation of the current user.

## B  Missing Proofs from Section 3

In this section, we provide the proofs missing from Section 3.

*Proof of Proposition 3.1.*  Consider a simple scenario where there are only two states $\{s, s'\}$ with $s$ being the initial state and two actions $\{x, x'\}$ that are feasible for both states. Without loss of generality, let $x$ be the action that the decision maker's algorithm chooses with probability at least $1/2$ at time $t = 1$. Now, consider an adversary that works in the following manner: It sets $f_t(s, \cdot) = 1$ and $f_t(s', \cdot) = 0$ for every $t \in [T]$. Regarding the state transition, the adversary sets $g_1(s, x) = s'$ and $g_1(s, x') = s$; and for every $t \in [2, T]$, it sets $g_t(s, \cdot) = s$ and $g_t(s', \cdot) = s'$. In such case, the

expected cumulative reward of the decision maker is at most $1 + T/2$, while the policy that always plays action $x_1$ obtains a cumulative reward of $T$. □

*Proof of Proposition 3.3.* Consider an ongoing chasing oracle that is implemented in a deterministic manner for a DRACC instance with $C = 1$, $W = 2$. The adversary chooses initial step $t_{\text{init}}$ and initial state $s_{\text{init}}$ so that $t_{\text{init}} = o(T)$, $|A_{t_{\text{init}}}| = 2$, and $\boldsymbol{\lambda}_{t_{\text{init}}} = \langle 0, 1 \rangle$. Note that throughout this proof, the inventory vectors and price vectors containing two elements are presented in an ordered way, which means that the first element corresponds to the resource with the smaller index.

The target policy $\gamma$ is chosen to have $\boldsymbol{\lambda}^{\gamma}_{t_{\text{init}}} = \langle 1, 1 \rangle$. Moreover, it maps every inventory vector to a price vector of $\langle \frac{2}{3}, \frac{1}{3} \rangle$. The adversary ensures the feasibility of such a policy by setting $t_e(i) = t$ for each resource that is sold out at $t$ with the price vector generated by $\gamma$, and setting $t_a(i') = t + 1$ for a new resource. With this setting, it holds for every $t \geq t_{\text{init}}$ that $\boldsymbol{\lambda}^{\gamma}_t = \langle 1, 1 \rangle$.

The adversary configures the valuation functions $v_t$ for each $t \geq t_{\text{init}}$ in an adaptive way, and ensures that for all such $t$

$$\hat{\boldsymbol{\lambda}}_t = \langle 0, 1 \rangle. \tag{6}$$

With the initial state $s_{\text{init}}$ chosen by the adversary, Eq. (6) holds for $t_{\text{init}}$. Suppose it holds for some $t \geq t_{\text{init}}$. Then the price vector $\hat{\boldsymbol{p}}$ generated by the oracle must be in the form of $\langle 1, p \rangle$ for some $p \in (0, 1]$. Let $i$ and $i'$ be the two resources in $A_t$ with $i < i'$. If $p \leq \frac{1}{3}$, the adversary sets $v_t(i) = \frac{2}{3}$ and $v_t(i') = \frac{1}{3}$. Then with the price vector generated by $\gamma$, payment $\frac{2}{3}$ is obtained from the user for resource $i$, while the oracle obtains payment $\frac{1}{3}$ from the user for $i'$. The difference in rewards is

$$f_t(\boldsymbol{\lambda}^{\gamma}_t, \boldsymbol{p}^{\gamma}_t) - f_t(\hat{\boldsymbol{\lambda}}_t, \hat{\boldsymbol{p}}_t) = \frac{1}{3}. \tag{7}$$

Moreover, since $i$ is sold out with $\boldsymbol{p}^{\gamma}_t$, the adversary sets $t_e(i) = t$ and $t_a(i'') = t + 1$ for a new resource $i'' > i'$. In such case, it is guaranteed that Eq. (6) holds for $t + 1$.

For the case where $p > \frac{1}{3}$, it can be verified that Eq. (7) still holds for $t$ and Eq. (6) holds for $t + 1$ when the adversary sets $v_t(i) = v_t(i') = \frac{1}{3}$. Since Eq. (7) is established for every $t \geq t_{\text{init}}$, $CR = \frac{1}{3}(t_{\text{final}} - t_{\text{init}})$. With $t_{\text{init}} = o(T)$, taking $t_{\text{final}} = T$ gives the desired bound. □

## C  Applications of the DRACC Problem

The mechanism LBPP proposed for the DRACC problem can be directly applied to a large family of online pricing problems arising in practice. Two examples are presented in this section: the *online job scheduling (OJS)* problem and the problem of *matching over dynamic bipartite graphs (MDBG)*.

### C.1  Online Job Scheduling

The OJS problem described in this section is motivated by the application of assigning jobs that arrive online to limited bandwidth slots for maximizing the total payments collected from the jobs. Formally, in the OJS problem, there are $T$ strategic myopic jobs, arriving sequentially over $N$ time slots. Each slot $i \in [N]$ lasts over the time interval $[i, i + 1)$ and is associated with a bandwidth $c(i)$, which means that this slot can be allocated to at most $c(i)$ jobs. For each job $t \in T$, the adversary specifies an arrival slot $1 \leq a_t \leq N$, a departure slot $a_t \leq d_t \leq N$, a length $1 \leq l_t \leq d_t - a_t + 1$, and a value $v_t \in [0, 1)$. We emphasize that any number (including zero) of jobs may have slot $i$ as their arrival (or departure) slot. The goal of job $t$ is to get an allocation of $l_t$ contiguous slots within $[a_t, d_t]$, namely, a slot interval in

$$\mathcal{I}_t = \{[i, i + l_t - 1] \mid a_t \leq i \leq d_t - l_t + 1\},$$

with $v_t$ being the job's value for each such allocation. Let $C$ and $W$ be upper bounds on $\max_{i \in [N]} c(i)$ and $\max_{t \in [T]} d_t - a_t + 1$, respectively.

Job $t \in [T]$ is reported to the OJS mechanism at the beginning of slot $a_t$; if several jobs share the same arrival slot, then they are reported to the mechanism sequentially in an arbitrary order. At the beginning of slot $a_t$, the mechanism is also informed of the bandwidth parameter $c(i)$ of every slot $i \in A_t$, where $A_t$ is defined to be the slot interval

$$A_t = [a_t, a_t + W - 1];$$

note that the mechanism may have been informed of the bandwidth parameters of some slots in $A_t$ beforehand (if they belong to $A_{t'}$ for $t' < t$). In response, the mechanism posts a price vector $\boldsymbol{p}_t \in (0,1]^{A_t}$ and elicits the parameters $d_t$, $l_t$, and $v_t$. Subsequently, (one bandwidth unit of) the slots in the demand set $\hat{A}_t^{\boldsymbol{p}_t}$ are allocated to job $t$ at a total price of $\hat{q}_t^{\boldsymbol{p}_t}$, where

$$
\hat{A}_t^{\boldsymbol{p}} = \begin{cases} \emptyset & \text{if } v_t < \min_{I \in \mathcal{I}_t} \sum_{i \in I} \boldsymbol{p}(i) \\ \operatorname{argmin}_{I \in \mathcal{I}_t} \sum_{i \in I} \boldsymbol{p}(i) & \text{otherwise} \end{cases} \qquad \text{and} \qquad \hat{q}_t^{\boldsymbol{p}} = \sum_{i \in \hat{A}_t^{\boldsymbol{p}}} \boldsymbol{p}(i)
$$

for any price vector $\boldsymbol{p} \in (0,1]^{A_t}$, consistently breaking $\operatorname{argmin}$ ties according to the lexicographic order on $A_t$.

Let $\boldsymbol{\lambda}_t \in \{0,1,\dots,C\}^{A_t}$ be the (remaining) bandwidth vector that encodes the number $\boldsymbol{\lambda}_t(i)$ of units remaining from the bandwidth of slot $i \in A_t$ before processing job $t = 1,\dots,T$. Formally, if slot $i$ has not been allocated to any of the jobs in $\{1,\dots,t-1\}$, then $\boldsymbol{\lambda}_t(i) = c(i)$; and if (a bandwidth unit of) slot $i$ is allocated to job $t$ and $i \in A_{t+1}$, then $\boldsymbol{\lambda}_{t+1}(i) = \boldsymbol{\lambda}_t(i) - 1$. We say that a price vector $\boldsymbol{p}$ is feasible for the bandwidth vector $\boldsymbol{\lambda}_t$ if $\boldsymbol{p}(i) = 1$ for every $i \in A_t$ such that $\boldsymbol{\lambda}_t(i) = 0$, that is, for every slot $i$ that has already been exhausted before job $t$ is processed. To ensure that the slots' bandwidth is not exceeded, we require that the posted price vector $\boldsymbol{p}_t$ is feasible for $\boldsymbol{\lambda}_t$ for every $1 \le t \le T$. We aim for posted price OJS mechanisms whose objective is to maximize the total expected payment $\mathbb{E}[\sum_{t=1}^T \hat{q}_t^{\boldsymbol{p}_t}]$ received from all jobs, where the expectation is over the mechanism's internal randomness.

A pricing policy $\gamma$ is a function that maps each bandwidth vector $\boldsymbol{\lambda} \in \{0,1,\dots,C\}^{A_t}$, $t \in [T]$, to a price vector $\boldsymbol{p} = \gamma(\boldsymbol{\lambda})$, subject to the constraint that $\boldsymbol{p}$ is feasible for $\boldsymbol{\lambda}$. Given a pricing policy $\gamma$, consider a decision maker that repeatedly plays according to $\gamma$; namely, she posts the price vector $\boldsymbol{p}_t^\gamma = \gamma(\boldsymbol{\lambda}_t^\gamma)$ for job $t = 1,\dots,T$, where $\boldsymbol{\lambda}_t^\gamma$ is the bandwidth vector obtained by applying $\gamma$ recursively on previous bandwidth vectors $\boldsymbol{\lambda}_{t'}^\gamma$ and posting prices $\gamma(\boldsymbol{\lambda}_{t'}^\gamma)$ for jobs $t' = 1,\dots,t-1$. Denoting $\hat{q}_t^\gamma = \hat{q}_t^{\boldsymbol{p}_t^\gamma}$, the revenue of this decision maker is given by $\sum_{t=1}^T \hat{q}_t^\gamma$. Given a collection $\Gamma$ of pricing policies, the quality of a posted price OJS mechanism $\{\boldsymbol{p}_t\}_{t=1}^T$ is measured by means of the decision maker's regret with respect to $\Gamma$, namely

$$
\max_{\gamma \in \Gamma} \sum_{t=1}^T \hat{q}_t^\gamma - \mathbb{E}\left[\sum_{t=1}^T \hat{q}_t^{\boldsymbol{p}_t}\right],
$$

where the expectation is taken over the decision maker's randomness.

**Reduction to DRACC.** Given the aforementioned choice of notation, the transformation of an OJS instance to a DRACC instance should now be straightforward. Specifically: job $t$ is mapped to user $t$; slot $i$ is mapped to resource $i$; slot $i$'s bandwidth parameter $c(i)$ is mapped to the capacity of resource $i$; job $t$'s arrival slot $a_t$ determines the set $A_t$ of active resources at time $t$, and through these sets, the arrival and departure times of the resources; and job $t$'s length $l_t$ and value $v_t$ parameters determine the valuation function of user $t$, assigning a value of $v_t$ to each $I \in \mathcal{I}_t$; and a zero value to any other subset of $A_t$. The following corollary is now inferred directly from Theorem 3.10.

**Corollary C.1.** *The OJS problem admits a mechanism whose regret for $T$-round instances is* $O\left(T^{\frac{1}{2}\left(1 + \frac{CW}{CW+1}\right)}\sqrt{\log |\Gamma|}\right)$.

Corollary C.1 is derived from the regret bound of LBPP for the DRACC problem with arbitrary valuation functions, based on the (randomized) chasing oracle implementation developed in Theorem 3.6. It turns out though that one can exploit the structural properties of the OJS problem to design a chasing oracle with dramatically improved chasing regret, thus improving the regret bound for the OJS problem (see Corollary C.3).

**Lemma C.2.** *The OJS problem admits a (deterministic) ongoing chasing oracle whose chasing regret is at most $2 \cdot CW$.*

*Proof.* One property of OJS is that for any two slots $i$ and $i'$,

$$
t_a(i) \le t_a(i') \quad \Rightarrow \quad t_e(i) \le t_e(i'). \tag{8}
$$

We prove the claim by constructing a chasing ongoing oracle $\mathcal{O}^{\texttt{Chasing}}$ with the desired CR using this property. Given a target policy $\gamma$, an initial step $t_{\text{init}}$, and an initial state $s_{\text{init}}$, oracle $\mathcal{O}^{\texttt{Chasing}}$ posts a price vector $\hat{\boldsymbol{p}}_t$ for each $t \geq t_{\text{init}}$ as follows

$$\hat{\boldsymbol{p}}_t = \begin{cases} \langle 1 \rangle_{i \in A_t} & \text{if } t \leq \min\{T, \max_{i \in A_{t_{\text{init}}}} t_e(i)\} \\ \boldsymbol{p}_t^{\gamma} & \text{otherwise} \end{cases} .$$

Let $t' = \min\{T, \max_{i \in A_{t_{\text{init}}}} t_e(i)\}$. The price vector $\hat{\boldsymbol{p}}_t$ is trivially feasible for every $t \in [t_{\text{init}}, t']$. If $t' < T$, then for every slot $i$ in $A_{t'+1}$, we have $i \notin A_{t_{\text{init}}}$, which gives $t_a(i) > t_{\text{init}}$. Since $\mathcal{O}^{\texttt{Chasing}}$ does not sell any slot to users from $t_{\text{init}}$ to $t'$, it holds that $\hat{\boldsymbol{\lambda}}_{t'+1}(i) = c(i) \geq \boldsymbol{\lambda}_{t'+1}^{\gamma}(i)$. Therefore, $\texttt{Good}_{t'+1} = A_{t'+1}$ and $\texttt{Bad}_{t'+1} = \emptyset$. Then can be proved inductively that for any $t \geq t' + 1$, $\texttt{Bad}_t = \emptyset$, which ensures the feasibility of $\hat{\boldsymbol{p}}_t$. Moreover, for each $t \geq t' + 1$, since $\hat{\boldsymbol{p}}_t = \boldsymbol{p}_t^{\gamma}$, we have $f_t(\hat{\boldsymbol{\lambda}}_t, \hat{\boldsymbol{p}}_t) = f_t(\boldsymbol{\lambda}_t^{\gamma}, \boldsymbol{p}_t^{\gamma})$.

It remains to bound $\sum_{t=t_{\text{init}}}^{t'} f_t(\boldsymbol{\lambda}_t^{\gamma}, \boldsymbol{p}_t^{\gamma}) - \sum_{t=t_{\text{init}}}^{t'} f_t(\hat{\boldsymbol{\lambda}}_t, \hat{\boldsymbol{p}}_t)$. Let $S$ be the set of slots $i$ with $t_a(i) \in (t_{\text{init}}, t']$. By Eq. (8), it holds for every $i \in S$ that $t_e(i) \geq t'$, because for every $i' \in A_{t_{\text{init}}}$, $t_a(i') \leq t_{\text{init}}$. By definition, $S \subseteq A_{t'}$, which gives $|S| \leq W$. Since the slots that can be sold by any policy to users in $[t_{\text{init}}, t']$ belong to $A_{t_{\text{init}}} \cup S$, we have

$$\sum_{t=t_{\text{init}}}^{t'} f_t(\boldsymbol{\lambda}_t^{\gamma}, \boldsymbol{p}_t^{\gamma}) \leq C \cdot |A_{t_{\text{init}}} \cup S| \leq C \cdot 2W .$$

Since $\sum_{t=t_{\text{init}}}^{t'} f_t(\hat{\boldsymbol{\lambda}}_t, \hat{\boldsymbol{p}}_t)$ is non-negative, this theorem is established. $\qquad\square$

By plugging Lemma C.2 into Theorem 3.8, we obtain the following improvement to Corollary C.1; this bound is near-optimal due to [9].

**Corollary C.3.** *The OJS problem admits a mechanism whose regret for $T$-round instances is* $O\left(\sqrt{CW \cdot T \log |\Gamma|}\right)$.

### C.2 Matching Over Dynamic Bipartite Graphs

The MDBG problem is a dynamic variation of the conventional bipartite matching problem with the goal of maximizing the revenue. Formally, in the MDBG problem, there are two sets of nodes, the *left-side* node set $\texttt{Left} = \{i\}_{i \in [N]}$ and the *right-side* node set $\texttt{Right} = \{t\}_{t \in [T]}$. The nodes in each of these two sets arrive sequentially and dynamically. For each node $i \in \texttt{Left}$, an adversary specifies a pair of parameters $t_a(i) \in [T]$ and $t_e(i) \in \left[t_a(i), [T]\right]$. It means that the node $i$ arrives just before the arrival of the node $t = t_a(i) \in \texttt{Right}$, and expires immediately after the node $t' = t_e(i) \in \texttt{Right}$ is given. For each node $t \in \texttt{Right}$, define $A_t = \{i \in \texttt{Left} : t \in [t_a(i), t_e(i)]\}$. The adversary also specifies a weight $w_t(i) \in [0, 1)$ for each $t \in \texttt{Right}$ and $i \in A_t$.

A posted price mechanism is required to present a price vector $\boldsymbol{p}_t \in (0, 1]^{|A_t|}$ independently of $w_t(\cdot)$ upon the arrival of each node $t \in \texttt{Right}$. For any price vector $\boldsymbol{p}$ presented to $t$, define $\hat{A}_t^{\boldsymbol{p}} = \text{argmax}_{i \in A_t} w_t(i) - \boldsymbol{p}(i)$ with breaking ties in a fixed way. The mechanism matches the left-side node $\hat{A}_t^{\boldsymbol{p}}$ to the right-side node $t$ and charges $t$ the payment $\boldsymbol{p}\left(\hat{A}_t^{\boldsymbol{p}}\right)$ if $w_t\left(\hat{A}_t^{\boldsymbol{p}}\right) \geq \boldsymbol{p}\left(\hat{A}_t^{\boldsymbol{p}}\right)$. Otherwise, no left node is matched to $t$, and no payment is obtained. After that, $w_t(\cdot)$ is revealed to the mechanism.

In the MDBG problem, every left-side node $i$ can only be matched to at most one right-side node $t$. We express this constraint as a feasibility requirement on the price vector that for each right-side node $t$, if a left-side node $i \in A_t$ has already been matched before the arrival of $t$, then the price of $i$ should be set to 1. The states of whether the left-side nodes in $A_t$ have been matched can be described with a Boolean vector of length $|A_t|$, and a pricing policy $\gamma \in \Gamma$ is a mapping from each possible Boolean vector to a feasible price vector. The objective of the MDBG problem is to find a feasible price vector $\boldsymbol{p}_t$ for every $t \in \texttt{Right}$ to maximize the total payments, and the regret is defined to be the difference between the revenue obtained by the best fixed in-hindsight pricing policy in a given collection $\Gamma$ and the expected revenue of the mechanism.

**Reduction to DRACC.** This problem can be transformed to a special case of the DRACC problem by taking the nodes in Left (resp. Right) as the resources (resp. users). The capacity of every resource is exactly one. The valuation function of each user $t$ maps each subset $A \subseteq A_t$ to $v_t(A) = \max_{i \in A} w_t(i)$. Such a setting is consistent because the price posted for each resource $i$ is strictly larger than 0, which ensures that at most one resource is allocated to each user. Moreover, $v_t$ is a $k_t$-demand valuation function with $k_t = 1$ for every $t$. Using Theorem 3.9, we get the following result.

**Corollary C.4.** *For the MDBG problem, the regret of the mechanism LBPP is bounded by* $O\left(W^{\frac{1}{4}}T^{\frac{3}{4}}\sqrt{\log|\Gamma|}\right)$.

# D External Regret

To complement our result, in this part we consider another natural alternative definition for regret, known as *external regret*, defined as follows (see [4] for more details).

$$\max_{\gamma \in \Gamma} \sum_{t \in [T]} f_t\Big(s_t, \gamma(s_t)\Big) - \sum_{t \in [T]} \mathbb{E}\Big[f_t(s_t, x_t)\Big]. \tag{9}$$

In words, while policy regret is the difference between the simulated reward of the optimal fixed policy and the actual reward of the algorithm, in external regret the reward that is being accredited to the optimal fixed policy in each round $t$ is the reward that policy would have obtained when being in the actual state of the algorithm (versus being in its simulated current state). In [4], it is shown that for the online learning problems where the reward functions depend on the $m$-recent actions, the policy regret and the external regret are incomparable, which means that any algorithm with a sublinear policy regret has a linear external regret, and vice visa. Based on the techniques proposed in [4], we prove that such a statement also holds for the online learning problem on the Dd-MDP with chasability. This is the reason why we focus on obtaining vanishing policy regret in the main part of this paper.

**Theorem D.1.** *There exists a $\sigma$-chasable instance of the Dd-MDP so that for any online learning algorithm having a sublinear policy regret on this instance, it cannot guarantee a sublinear external regret on the same instance, and vice visa.*

*Proof.* We start by constructing a deterministic MDP instance and proving that it is a feasible Dd-MDP instance with a constant $\sigma$.

$$\tag{10}$$

Consider the deterministic MDP instance with $m > 2$ states in Eq. (10), where $m$ is a constant independent of $T$. Each state in this instance is labeled with a distinct integer in $[m]$. The action set contains two actions, which are denoted by FORWARD and BACKWARD, respectively. These two actions are feasible for every state. The state transition functions $g_t$ and reward functions $f_t$ are fixed for all $t \in [T]$ as follows.

$$g_t(s, x) = \begin{cases} s+1 & \text{if } s < m \wedge x = \text{FORWARD} \\ m & \text{if } s = m \wedge x = \text{FORWARD} \\ 1 & \text{if } x = \text{BACKWARD} \end{cases},$$

$$f_t(s, x) = \begin{cases} 0.5 & \text{if } s = m \wedge x = \text{FORWARD} \\ 1 & \text{if } s = m \wedge x = \text{BACKWARD} \\ 0 & \text{otherwise} \end{cases}.$$

For any target policy $\gamma \in \Gamma$, initial time $t_{\text{init}}$ and any initial state $s_{\text{init}}$, let $k = m - s_{\text{init}}$, and $\{\hat{x}_t\}_{t \geq t_{\text{init}}}$ be a sequence of actions so that

$$\hat{x}_t = \begin{cases} \text{FORWARD} & \text{if } t \leq t_{\text{init}} + k - 1 \\ \text{BACKWARD} & \text{otherwise} \end{cases}.$$

This sequence of actions are trivially feasible. For any $\tau \leq t_{\mathrm{init}} + k - 1$, it is easy to see that

$$\sum_{t=t_{\mathrm{init}}}^{\tau} f_t(s^\gamma(t), x^\gamma(t)) - f_t(\hat{s}_t, \hat{x}_t) \leq m - 1,$$

where $\{\hat{s}_t\}_{t \geq t_{\mathrm{init}}}$ is a sequence of states defined in a similar with with Eq. (5). By the setting of the state transition function, we have $\hat{s}_{t_{\mathrm{init}}+k} = m$. Let $t' = \operatorname*{argmin}_{t \geq t_{\mathrm{init}}+k} x^\gamma(t) = \text{BACKWARD}$. Then

$$\sum_{t=t_{\mathrm{init}}+k}^{t'} f_t(s^\gamma(t), x^\gamma(t)) - f_t(\hat{s}_t, \hat{x}_t) \leq 0,$$

and for every $t > t'$,

$$f_t(s^\gamma(t), x^\gamma(t)) = f_t(\hat{s}_t, \hat{x}_t)$$

because in such a case $s^\gamma(t) = \hat{s}_t$ and $x^\gamma(t) = \hat{x}_t$ always hold. Putting the three formulas above together, it is proved that this Dd-MDP instance is $\sigma$-chasable with $\sigma = m - 1$.

Let the number of rounds that an arbitrary algorithm performs the actions FORWARD and BACKWARD at the state $m$ be $k$ and $k'$, respectively. Note that each time an algorithm performs BACKWARD at the state $m$, then it needs to take at least $m - 1$ rounds to go back to the state $m$. It implies that $k + m \cdot k' \leq T$. The total reward obtained by this algorithm is

$$\frac{1}{2}k + k' \leq \frac{1}{2}k + \frac{1}{m}(T - k). \tag{11}$$

Since the total reward by repeating a fixed policy $\gamma_F$ that maps every state to FORWARD is at least $\frac{1}{2}(T - m)$, the policy regret is at least $(\frac{1}{2} - \frac{1}{m})(T - k) - \frac{m}{2}$. Therefore, if the policy regret is sublinear in $T$, we have $k = T - o(T)$. Now, consider another policy $\gamma_B$ that maps every state to the action BACKWARD. We have

$$\sum_{t=1}^{T} f_t(s_t, \gamma'(s_t)) - \sum_{t=1}^{T} f_t(s_t, x_t) \geq \left(1 - \frac{1}{2}\right) \cdot k,$$

which implies that the external regret is linear in $T$.

Now consider an arbirary algorithm whose external regret is sublinear in $T$. Then, the total reward of this algorithm is at most $\frac{T}{m} + o(T)$, because otherwise it can still be inferred from Eq. (11) that $k$ is linear in $T$, which leads to a linear external regret. Recall that the total reward of repeating the policy $\gamma_F$ is $(T - m)/2$. Therefore, the policy regret is linear in $T$. □

# E Bandit Setting

In Section 3, we investigate the online learning problem on Dd-MDPs under the full information setting, which means that for each round $t$, both the state transition function $g_t(s, x)$ and reward function $f_t(s, x)$ selected by the adversary are completely revealed to the decision maker after the decision maker chooses a (randomized) action $x_t$. In this part, we consider the *bandit* setting, where at each round $t$, the decision maker only knows the actual reward she receives, $f_t(s_t, x_t)$, with the state transition function $g_t(\cdot, \cdot)$. Obtaining vanishing regret for Dd-MDPs under the bandit setting requires a stronger condition than $\sigma$-chasability, which is defined in the following subsection.

## E.1 Stateless Chasability

We say that an instance of Dd-MDP satisfies the *stateless chasability* condition for some parameter $\sigma > 0$ if there exists a chasing ongoing oracle $\mathcal{O}^{\text{Chasing}}$ which not only guarantees that $\text{CR} \leq \sigma$, but also ensures that for any target policy $\gamma$ and any initial state $t_{\mathrm{init}}$, the cumulative reward obtained by taking the generated actions $\{\hat{x}_t\}_{t \geq t_{\mathrm{init}}}$ does not depend on the initial state $s_{\mathrm{init}}$. More formally, let $s_{\mathrm{init}}, s'_{\mathrm{init}}$ be two arbitrary initial states, and $\{\hat{x}_t\}_{t \geq t_{\mathrm{init}}}, \{\hat{x}'_t\}_{t \geq t_{\mathrm{init}}}$ be two sequence of actions generated by the chasing ongoing oracle with starting from $s_{\mathrm{init}}$ and $s'_{\mathrm{init}}$, respectively. The stateless chasability condition requires that for any $t_{\mathrm{final}} \geq t_{\mathrm{init}}$

$$\sum_{t \in [t_{\mathrm{init}}, t_{\mathrm{final}}]} \mathbb{E}\big[f_t(\hat{s}_t, \hat{x}_t)\big] = \sum_{t \in [t_{\mathrm{init}}, t_{\mathrm{final}}]} \mathbb{E}\big[f_t(\hat{s}'_t, \hat{x}'_t)\big],$$

where $\hat{s}_t$ and $\hat{s}'_t$ are defined in a similar way with Eq. (5). A chasing ongoing oracle is said to be applicable to the bandit setting if its decision on each action $\hat{x}_t$ for $t \geq t_{\text{init}}$ only depends on $s_{\text{init}}$, $\{f_{t'}(\hat{s}_{t'}, \hat{x}_{t'})\}_{t' \in [t_{\text{init}}, t-1]}$ and $\{g_{t'}(\cdot, \cdot)\}_{t' \in [t_{\text{init}}, t-1]}$.

## E.2 Multiarmed Bandit Problem

To develop vanishing-regret algorithms for $\sigma$-chasable Dd-MDPs under the bandit setting, we utilize technical tools that are related to the *Multiarmed Bandit Problem (MBP)* [6]. Using a blackbox algorithm for this problem, Appendix E.3 shows how to obtain vanishing regret for our problem.

In MBP, there is a set of arms $\Gamma$, and $\Psi \in \mathbb{N}$ rounds. At each round $\psi \in [\Psi]$, an adversary specifies a reward function $F_\psi : \Gamma \mapsto [0, 1]$, which is unknown to the online algorithm at the beginning of this round. Simultaneously, the algorithm chooses an action $\gamma_\psi \in \Gamma$. Then the reward $F_\psi(\gamma_\psi)$ obtained by the algorithm is revealed. The goal of the algorithm is to pick a sequence of actions $\gamma_1, \ldots, \gamma_\Psi$ in an online fashion to maximize $\mathbb{E}\left[ \sum_{\psi \in [\Psi]} F_t(\gamma_\psi) \right]$. The regret is defined to be

$$\max_{\gamma \in \Gamma} \sum_{\psi \in [\Psi]} F_\psi(\gamma) - \sum_{\psi \in [\Psi]} \mathbb{E}\left[ F_\psi(\gamma_\psi) \right].$$

**Theorem E.1** ([5]). *There exists an algorithm* Implicitly Normalized Forecaster *(INF) for MBP whose regret is bounded by* $O\left( \sqrt{|\Gamma| \cdot \Psi} \right)$.

## E.3 Decision Making Algorithm: Chasing & Switching in Fixed-Length Periods

We now present our decision making (DM) algorithm for the $\sigma$-chasable Dd-MDP problems under the bandit setting. Our DM algorithm *Chasing and Switching in Fixed-Length Periods* (C&S-FLP) requires blackbox accesses to a chasing ongoing oracle that is applicable to the bandit setting and Algorithm INF for MBP, where the action set of MBP is set to be the collections of policies $\Gamma$ in Dd-MDP. Algorithm INF runs over consecutive periods of $\tau$ rounds for some $\tau > \sigma$, while the last period is allowed to have less than $\tau$ rounds. At the beginning of each period $\psi$, C&S-FLP invokes Algorithm INF to choose a policy $\gamma_\psi$ from $\Gamma$. Then it starts a new run of the chasing ongoing oracle $\mathcal{O}^{\text{Chasing}}$ with $\gamma_\psi$ as the target policy, $s_{(\psi-1)\tau+1}$ as the initial state $s_{\text{init}}$ and $(\psi-1)\tau + 1$ as the initial time $t_{\text{init}}$. Then C&S-FLP takes the sequence $\{\hat{x}_t\}_{t \in [(\psi-1)\tau+1, \min\{\psi \cdot \tau, T\}]}$ of actions generated by $\mathcal{O}^{\text{Chasing}}$ throughout the current period $\psi$, and send the reward $f_t(s_t, x_t)$ and state transition function $g_t(\cdot, \cdot)$ to $\mathcal{O}^{\text{Chasing}}$ after performing each $\hat{x}_t$ as the feedback. After the reward of the last step $t = \min\{T, \psi\tau\}$ of the current period is received, C&S-FLP computes $F_\psi(\gamma_\psi) = \frac{1}{\tau} \sum_{t'=(\psi-1)\tau+1}^{t} f_{t'}(s_{t'}, x_{t'})$ and feeds it to Algorithm INF as the reward of $\gamma$ at $t$.

**Theorem E.2.** *The regret of C&S-FLP is bounded by* $O\left( \frac{\sigma \cdot T}{\tau} + \sqrt{|\Gamma| T \tau} \right)$.

*Proof.* Let $\Psi = \lceil \frac{T}{\tau} \rceil$, and $R(\Psi, |\Gamma|)$ be the regret of Algorithm INF. With the stateless condition of the chasing ongoing oracle, $\{F_\psi\}_{\psi \in [\Psi]}$ is a sequence of stateless reward functions that satisfy the condition of Theorem E.2. Therefore, we have

$$\max_{\gamma \in \Gamma} \sum_{\psi \in [\Psi]} F_\psi(\gamma) - \sum_{\psi \in [\Psi]} \mathbb{E}\left[ F_\psi(\gamma_\psi) \right] \leq R(\Psi, |\Gamma|),$$

which gives that

$$\tau \cdot \max_{\gamma \in \Gamma} \sum_{\psi \in [\Psi]} F_\psi(\gamma) - \sum_{t \in [T]} \mathbb{E}\left[ f_t(s_t, x_t) \right] \leq \tau \cdot R(\Psi, |\Gamma|).$$

By the definition of CR, for each period $\psi$ we have

$$\sum_{t=(\psi-1)\tau+1}^{\psi \cdot \tau} f_t(s^\gamma(t), x^\gamma(t)) - \tau \cdot \max_{\gamma \in \Gamma} \sum_{\psi \in [\Psi]} F_\psi(\gamma) \leq \sigma.$$

Therefore, the regret of Algorithm C&S-FLP is bounded by

$$\sum_{t \in [T]} f_t(s^\gamma(t), x^\gamma(t)) - \sum_{t \in [T]} \mathbb{E}\Big[f_t(s_t, x_t)\Big] \le \sigma \cdot \Psi + \tau \cdot R(\Psi, |\Gamma|) \,.$$

This proposition is proved by plugging Theorem E.1 into the formula above. □

**Corollary E.3.** *By taking $\tau = T^{\frac{1}{3}}$, the regret of Algorithm C&S-FLP is bounded by $O\left(\sigma T^{\frac{2}{3}}\sqrt{|\Gamma|}\right)$.*

# F  Lower Bounds for Online Learning over Dd-MDPs with Chasability

In this part, we will prove lower bounds on the regret of online learning algorithms for $\sigma$-chasable Dd-MDP instances under the full-information setting and bandit setting, respectively.

**Theorem F.1.** *The regret of any online learning algorithm for $1$-chasable Dd-MDP under full-information (resp., bandit) feedback is lower bounded by $\Omega(\sqrt{T \log |\Gamma|})$ (resp., $\Omega(|\Gamma|^{1/3}T^{2/3})$).*

*Proof.* We only prove the full information lower bound, and the proof of the bandit version follows the same lines. Suppose the statement of the theorem does not hold. Then, there exists an online learning algorithm with regret of $o(\sqrt{T \log |\Gamma|})$ for any $1$-chasable instance of the Dd-MDP problem. We show how to use this algorithm to design an OLSC algorithm with a unit switching cost whose regret is $o(\sqrt{T \log |\mathcal{X}|})$, where $\mathcal{X}$ is the action set of the OLSC instance. This is in contradiction to the known information theoretic $\Omega(\sqrt{T \log |\mathcal{X}|})$ lower bound on the regret of OLSC under the full-information setting [13, 24, 31]. (For the bandit version of this proof we use the lower bound of [18]).

Here is how the reduction works: Given an OLSC instance with a set $\mathcal{X}$ of actions and a unit switching cost, we construct a Dd-MDP instance with a state $s^x$ for each action $x \in \mathcal{X}$. An arbitrary state $s \in S$ is selected to be the initial state $s_1$. Moreover, we set $\mathcal{X}_s = \mathcal{X}$ for every state $s$. For every $x \in \mathcal{X}$, we introduce a policy $\gamma^x$ in the policy collection $\Gamma$ of the Dd-MDP, defined so that it maps all states to action $x \in \mathcal{X}$. For each round $t$, when the adversary in OLSC specifies a reward function $F_t(\cdot)$, we construct the state transition function and reward function in the Dd-MDP by setting

$$g_t(s, x) = s^x \qquad \text{and} \qquad f_t(s, x) = \frac{1}{2}F_t(x) + \frac{1}{2} \cdot 1_{s=s^x} \,.$$

Obviously, this is a $1$-chasable Dd-MDP instance. Moreover,

$$\max_{\gamma^x \in \Gamma} \sum_{t \in [T]} f_t(s^{\gamma^x}(t), x^{\gamma^x}(t)) - \sum_{t \in [T]} \mathbb{E}\Big[f_t(s_t, x_t)\Big] \le o(\sqrt{T \log |\Gamma|}) = o(\sqrt{T \log |\mathcal{X}|}) \,,$$

where the inequality is due to the assumed regret bound. The construction of the Dd-MDP instance ensures that for every $\gamma \in \Gamma$,

$$\sum_{t \in [T]} f_t(s^{\gamma^x}(t), x^{\gamma^x}(t)) \ge \frac{1}{2} \cdot \sum_{t \in [T]} F_t(x) + \frac{1}{2}(T - 1) \,,$$

$$\sum_{t \in [T]} f_t(s_t, x_t) \le \frac{1}{2} \sum_{t \in [T]} F_t(x_t) + \frac{1}{2}T - \frac{1}{2} \sum_{t \in [2,T]} 1_{x_t \ne x_{t-1}}$$

Putting these pieces together, we get

$$\frac{1}{2}\left( \max_{x \in \mathcal{X}} \sum_{t \in [T]} F_t(x) - \sum_{t \in [T]} \mathbb{E}\Big[F_t(x_t)\Big] + \sum_{t \in [2,T]} 1_{x_t \ne x_{t-1}} \right) - 1 \le o\left(\sqrt{T \log |\mathcal{X}|}\right) \,,$$

and therefore the regret of the OLSC instance is bounded by $o(\sqrt{T \log |\mathcal{X}|})$, a contradiction. □

# G Vanishing Regret and the DRACC Parameters

In this section, we prove that the vanishing regret is impossible if $C \cdot W$ grows linearly with $T$.

**Theorem G.1.** *If $C \cdot W = \Omega(T)$, then the regret of any posted price mechanism is $\Omega(T)$.*

*Proof.* Here we construct two instances of DRACC. The following settings are the same between these two instances.

- The parameters $C$ and $W$ are chosen so that $C \cdot W = \frac{T}{2}$. Set $N = W$.

- For each resource $i$, $t_a(i) = 1$ and $t_e(i) = T$. This setting implies that for every user $t$, $A_t = [N]$, which is consistent with $W = N$. Every $i \in [N]$ has the same capacity $c(i) = C$.

- For each user $t \in \left[1, \frac{T}{2}\right]$, the valuation function $v_t$ is set as follows.

$$
v_t(A') = \begin{cases} \frac{1}{2} & \text{if } |A'| = 1 \\ 0 & \text{otherwise} \end{cases} \quad \forall A' \subseteq A_t \,.
$$

For the users $t \in \left[\frac{T}{2} + 1, T\right]$, their valuation functions are different between the two instances. In particular, in the first instance, $v_t(A') = 0$ for any $A' \subseteq A_t$, while in the second instance

$$
v_t(A') = \begin{cases} 1 - \epsilon & \text{if } |A'| = 1 \\ 0 & \text{otherwise} \end{cases} \quad \forall A' \subseteq A_t \,.
$$

where $\epsilon$ is some small enough constant in $(0, \frac{1}{2})$.

Now consider an arbitrary deterministic mechanism $\mathcal{M}$. Such a mechanism will output the same sequence of price vectors for the first half of the users in these two instances. Therefore, the total number of resources that are allocated by $\mathcal{M}$ to the first half of users must be the same $k$ in the two instances for some $k \in \left[0, \frac{T}{2}\right]$. Then, the revenue of $\mathcal{M}$ is at most $\frac{k}{2}$ in the former instance, while at most $\frac{k}{2} + \left(\frac{T}{2} - k\right) \cdot (1 - \epsilon) = \frac{1-\epsilon}{2}T - (\frac{1}{2} - \epsilon)k$ in the latter one. Now consider a pricing policy $\gamma$ that maps every inventory vector except $\langle 0 \rangle$ to a price vector that only contains $\frac{1}{2}$. The revenue of $\gamma$ in the first instance is $\frac{T}{4}$. Similarly, there exists a policy $\gamma'$ with revenue $\frac{T}{2} \cdot (1 - \epsilon)$ in the second instance. Therefore, the regret of $\mathcal{M}$ is at least

$$
\max\left\{\frac{T}{4} - \frac{k}{2}, \; \frac{T}{2}(1 - \epsilon) - \left[\frac{1-\epsilon}{2}T - (\frac{1}{2} - \epsilon)k\right]\right\} \geq \frac{1 - 2\epsilon}{8(1 - \epsilon)}T \,.
$$

To generalize the result above to the mechanisms that can utilize the random bits, here we adopt Yao's principle [33]. In particular, we construct a distribution over the inputs which assigns probabilities $\frac{1-2\epsilon}{2-2\epsilon}$ and $\frac{1}{2-2\epsilon}$ to the two instances constructed above, respectively. It can be verified that against such a distribution, the expectation of any random mechanism's regret is at least $\frac{1-2\epsilon}{8(1-\epsilon)}T$. By Yao's principle, the lower bound on the regret of any mechanism that can utilizes the random bits is also $\frac{1-2\epsilon}{8(1-\epsilon)}T$. Therefore, this proposition is established. $\square$