[Reviews · NeurIPS 2020]

Review 1

Summary and Contributions: This paper considers a dynamic resource pricing problem. Agents arrive over time requesting resources, and the resources themselves become available and unavailable over time. The system sets (dynamic) prices on resources at each moment in time, and agents then choose the resources that maximize their utility given the prices. The agent requests are adversarial, and the goal is to select a pricing policy minimizes regret. The main contribution is a policy with vanishing regret for a very general formulation of such allocation problems. The paper frames the problem as an even more general dynamic MDP, which does not admit vanishing-regret policies in general. The main conceptual contribution is to show that if the dynamic MDP admits a "chasing oracle," then a vanishing-regret policy is possible (and they find a policy with asymptotically optimal regret). Essentially, the property required is that, starting from any system state, it is possible to simulate a policy that requires a *different* initial state, while suffering only a bounded amount of regret. We can think of this as the cost of "fixing" the initial state, e.g. by "chasing" the sequence of states that would have occurred had we started at the "correct" initial state. The authors then complete the result by showing that their class of resource allocation problems admit such chasing oracles.

Strengths: I like this paper a lot. Dynamic pricing for resource allocation is a central problem in the intersection of economics and computation/ML, with slow and steady progress over the last few years. This paper presents a very general solution. Moreover, the paper has a helpful conceptual message: it distills out the "difficult core" of the general learning problem (chasing a desired system state), and shows that this issue can be overcome in general resource allocation problems. I think this latter conceptual contribution is at least as valuable as the learning result itself! The technical results are highly non-trivial. I did not verify the details, but the authors do a good job of describing their high-level approach in the main body of the paper and I don't see any fatal flaws in the methodology. I appreciated the care taken to encode multiple different classic problems in their general setup.

Weaknesses: The generality of the setup makes the paper hard to read without significant effort. It took a few reads for me to fully internalize what the "chasability" property is capturing, despite being very familiar with many of the motivating applications. The encodings of the motivating problems are also not very straightforward. I think the paper could have been written more clearly. But this is a minor concern.

Correctness: As far as I understand, the claims and method are correct. (Though I did not verify all details of proofs.)

Clarity: The exposition is generally clear. The generality of the model makes the paper difficult to follow in places; I might have recommended having a running example that is simpler, then generalizing afterwards.

Relation to Prior Work: The relationship to prior work is clear.

Reproducibility: Yes

Additional Feedback: Post-rebuttal: I am still positive about the paper. Given the suggestions in the other reviews and the rebuttal, I'm hopeful that the authors can address some of the readability concerns in a camera-ready revision.


Review 2

Summary and Contributions: The paper considers a general set of online decisionmaking problems, with a goal of solving a dynamic pricing problem. The most general is dynamic deterministic Markov decision processes (Dd-MDP), where it is shown that there are no-regret policies if the setting satisfies a natural "chaseable" condition. From there, one obtains no-regret for dynamic resource allocation with capacity constraints (DRACC). This is the main goal of the paper: setting prices so as to allocate scarce, dynamically-arriving resources in an online way. Buyers have some valuation function over subsets of items (but want at most one of each item). Better bounds are given for nicer classes of valuation functions. The "chaseable" condition is roughly that, given a policy gamma, there is an algorithm that can have vanishing regret to gamma, even if it is started in a different state than gamma. To get no regret for Dd-MDP, there is an interesting use of algorithms for no-regret learning with switching costs. The idea is to "chase" the current policy of that algorithm, which cannot switch too often.

Strengths: I am slightly skeptical about the practical relevance of the DRACC problem to actual online pricing (though I could be wrong). But I think this question is interesting and relevant to researchers in the NeurIPS community, and the techniques and Dd-MDP problem I found very nice. The conceptual and technical contributions both seem valuable. I am not expert enough to assert this approach is novel, but it is as far as I know.

Weaknesses: No noticeable weaknesses in my opinion.

Correctness: Although I did not check all proofs, I gained confidence from reading the paper that the claims and proofs are correct.

Clarity: Yes, I thought so. I liked the organization.

Relation to Prior Work: Yes, I think so.

Reproducibility: Yes

Additional Feedback: The bandit version of Dd-MDP was mentioned a couple times. I wanted to clarify that this is investigated for its own benefit, but doesn't impact the analysis of DRACC (which just relies on the full-feedback version). Is that right? -------- After author response: My opinion is still positive. A common theme for reviewers is that the paper is hard to access and dense. I don't have any concrete suggestions unfortunately, but maybe keeping this in mind can help with revising.


Review 3

Summary and Contributions: In this paper, the authors consider a problem of dynamic resource allocation with capacity constraints. They first show that the problem is in general not solvable under the regret framework, and then show that how to bypass this difficulty with a notion of chasability.

Strengths: The setting captures a broad spectrum of applications, and the authors have proposed provable solutions for it.

Weaknesses: The interpretation of the chasability condition is not quite clear to me. What does this mean in the context of the applications mentioned? It would also be helpful if the authors could state the key novelty in techniques (aside from the extra chasability assumption and the reduction).

Correctness: Seems to be correct.

Clarity: Yes

Relation to Prior Work: Yes

Reproducibility: Yes

Additional Feedback:

[Author Response · NeurIPS 2020]

We would like to thank all the reviewers for their thoughtful and helpful comments. We address these comments for each reviewer separately below.

**Response to R1:**

Thanks for your helpful suggestions and thoughtful read of the paper. Here is how we intend to incorporate them to improve the writing:

- We agree with you that while the generality of our setup makes it possible to include all the applications in one framework, our setup can still benefit from further clarifications. In fact, exactly for the reason you mentioned, we wrote our analysis for the DRACC problem in separate parts and in the style of a running example. Nevertheless, things can be clarified further, especially with the interpretations of the chasability condition, and we intend to do so in the camera ready version if the paper is accepted (see also the second bullet in our response to reviewer R3).

- Based on your suggestion, we will revise the sections presenting the applications and their encoding as DRACC instances; we intend to add non-technical explanations to make the connections clearer.

**Response to R2:**

Thanks for your careful read and comments.

- Regarding your question on the bandit feedback setting, you are right: this setting is investigated for its own benefit and does not affect the analysis of the DRACC problem.

**Response to R3:**

Thanks for your review and questions. Let us try to address them briefly:

- While chasability is an assumption in the context of Dd-MDP, we would like to emphasize that it is not an assumption in the DRACC problem (and the applications derived from it), but rather a condition that this problem satisfies — as we prove in Section 3. The DRACC problem is the main motivating application behind our study, whereas the Dd-MDP setting is an abstraction that facilitates putting the different technical ingredients in the right scope. We believe that this abstraction can potentially find other applications for which the chasability condition is satisfied.

- In terms of the DRACC application, chasability means that the online algorithm can "simulate" a given pricing policy, while incurring a small revenue loss, even if the online algorithm starts from a (coordinate-wise) smaller inventory vector — this point will be clarified in the camera ready version if the paper is accepted. This requires a careful choice of price vectors because once the inventory of a resource is exhausted, the algorithm must post a unit price (which means that the resource cannot be purchased) — see the proofs of Theorems 3.4 and 3.6 for the technical details of how this is done.

- Prior work on online resource allocation was not able to achieve vanishing regret for settings as general as the DRACC problem since it did not handle the statefulness aspect of the problem. The key novelty in our paper is therefore introducing the Dd-MDP framework together with the chasability condition, showing that on the one hand, it suffices to ensure vanishing regret, and on the other hand, it is satisfied by natural online stateful price posting settings such as DRACC.

**Regarding the broader impact concerns (item 11 in the reviews)**:

The NeurIPS 2020 FAQ for Authors document mentions that *In general, if a paper presents theoretical work without any foreseeable impact in the society, authors can simply state "This work does not present any foreseeable societal consequence"*. Given the theoretical nature of our paper, we simply intended to follow this guideline.

[Meta-Review · NeurIPS 2020]

The reviewers evaluated the paper and there was broad agreement that this is quality work. However, there was still some concerns about readability so we would urge you to revise the submission for the camera-ready.